# Appraisal of published guidelines in European countries addressing the clinical care of childhood sexual abuse: protocol for a systematic review

Gabriel Otterman ,[1] Ulugbek Nurmatov,[2] Ather Akhlaq ,[3] Aideen Naughton,[4] Alison Mary Kemp ,[2] Laura Korhonen ,[1] Andreas Jud ,[5] Mary Jo Vollmer Sandholm,[6] Eva Mora-Theuer,[7] Sarah Moultrie,[8] Martin Chalumeau,[9] Wouter A Karst,[10] Jordan Greenbaum[11]

**Correspondence to**
Dr Gabriel Otterman;
gabriel.otterman@liu.se

## ABSTRACT

**Introduction** Childhood sexual abuse (CSA) is a global public health problem with potentially severe health and mental health consequences. Healthcare professionals (HCPs) should be familiar with risk factors and potential indicators of CSA, and able to provide appropriate medical management. The WHO issued global guidelines for the clinical care of children with CSA, based on rigorous review of the evidence base. The current systematic review identifies existing CSA guidelines issued by government agencies and academic societies in the European Region and assesses their quality and clarity to illuminate strengths and identify opportunities for improvement.

**Methods and analysis** This 10-database systematic review will be conducted according to the Centre for Reviews and Dissemination guidelines and will be reported according to The Preferred Reporting Items for Systematic Reviews and Meta-Analyses statement. Guidance for HCPs regarding CSA, written by a national governmental agency or academic society of HCPs within 34 COST Action 19106 Network Countries (CANC) and published in peer-reviewed or grey literature between January 2012 and November 2022, is eligible for inclusion. Two independent researchers will search the international literature, screen, review and extract data. Included guidelines will be assessed for completeness and clarity, compared with the WHO 2017/2019 guidelines on CSA, and evaluated for consistency between the CANC guidelines. The Appraisal of Guidelines for Research and Evaluation II tool and Grading of Recommendations Assessment, Development and Evaluation methodology will be used to evaluate CANC guidelines. Descriptive statistics will summarise content similarities and differences between the WHO guidelines and national guidelines; data will be summarised using counts, frequencies, proportions and per cent agreement between country-specific guidelines and the WHO 2017/2019 guidelines.

**Ethics and dissemination** There are no individuals or protected health information involved and no safety issues

## STRENGTHS AND LIMITATIONS OF THIS STUDY

⇒ This systematic review will be the first in this field of research ever undertaken that evaluates the quality and consistency of child sexual abuse (CSA) clinical guidelines across European countries.

⇒ An international team of subject-matter experts in the field of child maltreatment will conduct this study with experts in systematic review methodology using a robust and transparent approach.

⇒ For the comparison of CSA clinical guidelines in COST Action 19106 Network Countries, a reference standard—the WHO evidence-based guidelines—will be employed.

⇒ The different methods used in developing the guidelines across a diversity of practice settings may underpin the variability of both scope and quality of guidelines.

⇒ Our search strategy that is limited in scope to a national level may miss guidelines that are used in practice in European countries where healthcare is organised at the state, provincial or municipal level.

identified. Results will be published in a peer-reviewed medical journal.

**PROSPERO registration number** CRD42022320747.

## INTRODUCTION
### Rationale

Childhood sexual abuse (CSA), as defined by the United Nations and the WHO, encompasses a spectrum of inappropriate and (potentially) criminal sexual activities involving children and youth under 18 years of age.[1 2] Rates of CSA are generally high across the globe, with prevalence estimates ranging from 3.7% to 40% in European countries.[3] Adverse physical health, mental health and societal consequences of CSA include infection (eg, HIV, syphilis and other sexually transmitted infections); anogenital injury; unwanted pregnancy; substance

abuse disorders; post-traumatic stress disorder; major depression, often associated with self-harm and suicidal ideation; anxiety disorders and behavioural challenges, including antisocial behaviour.[4 5] As a result of these health conditions, many children and youth who experience one or more forms of CSA seek care from health professionals. They may present for acute care after a specific event or may seek care for other reasons which may or may not be related to their victimisation. In the latter case, patients may not disclose their status and the healthcare professional (HCP) must be alert to potential indicators and risk factors of CSA such as extra vulnerabilities to victimisation and inequitable access to care among children with disabilities.[6 7] In addition, the clinician must be knowledgeable about the appropriate set of responses in the care and follow-up of CSA and skilled at providing care using a trauma-informed, patient-centred approach.[8] Training of HCPs on CSA is often lacking in Europe.[9–12] Similarly, medical and nursing students are not routinely taught about trauma-informed, patient-centred care, which involves recognising the potential impact of prior traumatising events on a patient, caregiver and others, and responding with empathy and support. With a trauma-informed approach, the HCP makes every effort to minimise patient and caregiver distress during the visit, to actively involve these individuals in discussions and decisions, to prioritise their needs and facilitate short-term and long-term healing.[13]

To assist HCPs in recognising and appropriately responding to CSA, healthcare organisations need specific protocols and guidelines providing detailed information that aims to optimise care and maximise efficiency. Based on rigorous review of the global scientific literature on childhood sexual violence,[14] the WHO published evidence-based, comprehensive clinical practice guidelines (CPGs) in 2017 and 2019.[2 15] In addition, within the reference CPGs, good practice statements on clinical management are included as guidance even when there is a lack of evidence to support specific recommendations. Countries may rely on these documents to guide their HCPs or base their national guidelines on WHO content. National and regional medical academic societies also publish guidelines.[16–20] However, the nature of CSA guidelines varies in content and detail. HCPs in many countries do not have access to guidelines addressing CSA, due to lack of resources at the local, national or geographical regional level and lack of awareness of the WHO guidelines. This may lead to differences across countries and regions in access to evidence-based medical care for vulnerable children and youth.

It is important to identify existing CSA guidelines issued by government agencies and academic societies in the European Region and assess their quality and clarity in order to illuminate strengths and identify opportunities for improvement. This will help drive a targeted approach to enhancing the healthcare recognition of and response to CSA and facilitate comprehensive protocol development across the European Region. We posit that disparities in healthcare and health-related outcomes for survivors will decrease as quality of care is increased across the European countries.

## Objectives

The aim of this systematic review is to critically appraise existing clinical guidelines addressing the clinical care of CSA published in the COST Action 19106 Network Countries (CANC).[21] The review will address the following questions:

1. In comparison to the WHO's 2017 and 2019 clinical guidelines on the health sector recognition of and response to CSA, what are the completeness, clarity and consistency of healthcare guidelines published by countries within the European Region addressing CSA?
2. How do the included CANC countries' guidelines compare with one another in content consistency, and what discrepancies are manifest in the comparison between countries?

## METHODS

This systematic review will be conducted according to the Centre for Reviews and Dissemination guidelines and will be reported according to guidance from the Preferred Reporting Items for Systematic Reviews and Meta-Analyses (PRISMA) statement.[22]

## Registration

In accordance with PRISMA-Protocol guidelines, our systematic review protocol was registered with the International Prospective Register of Systematic Reviews (PROSPERO)[23] on registration number: (CRD42022320747).

## Eligibility criteria

Guidelines will be eligible for inclusion in the systematic review if they meet all the following criteria:

► Include specific guidance—that may include good practice statements—for HCPs to recognise and appropriately respond to children (<18 years) who may have experienced CSA.
► Are published in peer-reviewed journals or in the grey literature (including government or health professional society websites).
► Represent the product of a national governmental agency or an academic society of health professionals located within the CANC.
► Are published between January 2012 and November 2022 (the time period of 10 years was determined based on the subject-matter experts' familiarity as clinicians with the publication and utilisation of CSA guidelines, and the estimated frequency of updates of guidelines required to maintain their validity in the face of developments in the underlying published evidence).[24] Reports issued prior to this time may be outdated in certain topics such as diagnosis and treatment of infection. In addition, the beginning of the study period in January 2012 coincides with the

introduction of significant legislation in the European Union in December 2011 of Directive 2011/93—Combating the sexual abuse and sexual exploitation of children and child pornography, which specifies responsibilities of member states to ensure support of victims of CSA.[25]

► No language restrictions.

## Information sources
### Literature search
The systematic review will entail a search of CINAHL, EMBASE, Google Scholar, Guidelines International Network, MEDLINE, PsycINFO, SUMSearch, TRIP, Web of Science and WHO Global Health Library. References of the guidelines included in the review will be searched for additional guidelines. Search of the grey literature will include the Health Management Information Consortium, the National Institute for Health and Care Excellence (UK), as well as the websites of government health agencies and academic societies within the European Region. In addition, members of the Multi-sectoral Responses to Child Abuse and Neglect in Europe (COST Action 19106, a network of experts in child maltreatment and relevant stakeholders representing 34 countries) will be contacted and asked to identify additional relevant guidelines. Paediatricians, forensic physicians, gynaecologists and other physicians in the CANC engaged in the care of vulnerable children will be included in the search using a snowball strategy.

### Search strategy
Search terms will be developed under the expert guidance of one of the research team leads (UN) using medical subject headings (MeSH) and text words related to forms of childhood sexual abuse, and healthcare professional guidance. A draft of a MEDLINE search strategy is included in online supplemental appendix 1. After finalisation of the MEDLINE search strategy criteria, the latter will be adapted for the remaining electronic databases. The time period and language restrictions for guideline inclusion are described above.

## Patient and public involvement statement
Patients and the public were not involved in this systematic review of published national guidelines.

## Study period for the systematic review
The study commenced in May 2022 and completion is anticipated in May 2023.

## Study records
### Data management
Retrieved studies from all international electronic databases will be imported into EndNote software and deduplicated, before being screened and identified as eligible guidelines. Completed screening records, and full texts of included studies will be stored as a Master Database in EndNote software for evaluation.

### Selection process
Two authors will search databases and screen titles and abstracts independently for potentially eligible studies. Disagreement between researchers will be resolved by consensus or arbitration involving a third author where necessary. Full texts of studies will be retrieved for selected guidelines, and two authors will evaluate whether these meet inclusion/exclusion criteria. Disagreement will be resolved by referral to a third author if necessary. Reasons for the excluded papers will also be provided. The list of final selected studies will be shared with experts (maximum 3) for any missing or ongoing/unpublished studies.

### Data collection process
Two independent reviewers will extract data from included guidelines and enter it into a customised data extraction sheet. The two sheets will be compared for consistency and discrepancies resolved as above. The resultant data will be entered into the master database (Microsoft Excel). Non-English guidelines will be translated using readily available tools such as Google Translate and by consulting individuals who are bilingual, including in that particular language.

### Data items
The customised data extraction sheet will include the following information: country, year of dissemination, guideline development process with the report of the group membership involved, search method, and specific content and characteristics of guidelines as they relate to the good practice statements and recommendations from the 'Responding to children and adolescents who have been sexually abused: WHO 2017 clinical guidelines'[2] and the sections on identification of CSA included in the 'WHO Guidelines for the Health Sector Response to Child Maltreatment: Technical Report, 2019.'[15] Examples of content and characteristics to be assessed and extracted include components of trauma-informed care and of HIV postexposure prophylaxis.

### Outcomes and prioritisation
Primary outcome(s):
   Alignment with WHO guidelines.[2 15] Completeness, clarity and consistency of national guidelines compared with reference standard.
   Secondary outcomes:
   Consistency between the included clinical guidelines of the CANC.

### Risk of bias in individual studies
Eligible guidelines will be critically appraised by two or more independent reviewers working in duplicate. The quality, methodological rigour and transparency of included clinical guidelines will be assessed using Appraisal of Guidelines for Research and Evaluation II tool.[26 27]

## Data synthesis

Descriptive statistics will be employed to characterise content similarities and differences between the reference standards 'Responding to children and adolescents who have been sexually abused: WHO 2017 clinical guidelines'[2] and 'WHO guidelines for the health sector response to child maltreatment: Technical report, 2019'[15] and included national guidelines for the recognition, diagnosis of and response to CSA issued in European countries. Key findings from each guideline will be summarised and presented in descriptive summary tables. Data will be reported using counts, frequencies and proportions (eg, the proportion of guidelines that address a trauma-informed approach). Members of the research team will extract key quality indicators from the WHO 2017/2019 guidelines. Each country-specific guideline will be examined for the presence of the indicators. The per cent agreement on the rating of country-specific guidelines will be calculated and presented. In a subgroup analysis, we will compare identified guidelines published in the period from 2012 that preceded publication of the reference WHO guideline in 2017 to those published subsequently. We will summarise the findings by providing a systematic narrative synthesis with the information presented in text and tables.

## Meta-Biases

No assessment for meta-bias is planned for this protocol review.

## Confidence in cumulative estimate

The Grading of Recommendations Assessment, Development and Evaluation working group methodology will be used to assess the clarity and strength of underlying evidence inside guideline statements. In addition, the scope description and grade or rating of the guideline recommendations will be assessed.[28]

## Ethics and dissemination

The systematic review does not involve individuals or protected health information and there are no safety issues identified. Dissemination of study results will be accomplished through publication in a peer-reviewed medical journal.

### Author affiliations
[1]Barnafrid Centre, Department of Biomedical and Clinical Sciences, Linköping University, Linköping, Sweden
[2]Division of Population Medicine, School of Medicine, Cardiff University, Cardiff, UK
[3]Department of Health and Hospital Management, Institute of Business Management, Karachi, Pakistan
[4]National Safeguarding Service, Public Health Wales (NHS), Cardiff, UK
[5]Clinic for Child and Adolescent Psychiatry/Psychotherapy, University Hospital Ulm, Ulm, Germany
[6]Department of Forensic Sciences, Oslo University Hospital, Oslo, Norway
[7]Department of Paediatrics and Adolescent Medicine, Medical University of Vienna, Vienna, Austria
[8]Pediatric Trauma, UCSF Benioff Children's Hospital, Oakland, California, USA
[9]Department of General Pediatrics and Pediatric Infectious Diseases, Necker-Enfants Malades Hospitals, Paris, France
[10]Forensic Medicine, GGD Branbant Zuidoost, Eindhoven, Netherlands
[11]International Centre for Missing & Exploited Children, Atlanta, Georgia, USA

**Acknowledgements** The authors would like to acknowledge the International Centre for Missing & Exploited Children (ICMEC) for their valuable administrative support in the development of this protocol. We would also like to acknowledge the generous support of the COST Association in the establishment and maintenance of the Euro-CAN COST Action 19106 network.

**Contributors** GO and JG conceived the idea for this study. UN, GO and JG developed the methods and together with AA, AN, AMK, LK, AJ, MJVS, EM-T, SM, MC and WAK drafted this protocol.

**Funding** This work was supported by the International Centre for Missing and Exploited Children (no specific grant number).

**Competing interests** None declared.

**Patient and public involvement** Patients and/or the public were not involved in the design, or conduct, or reporting, or dissemination plans of this research.

**Patient consent for publication** Not applicable.

**Provenance and peer review** Not commissioned; externally peer reviewed.

**ORCID iDs**
Gabriel Otterman http://orcid.org/0000-0003-4837-9614
Ather Akhlaq http://orcid.org/0000-0001-6210-4583
Alison Mary Kemp http://orcid.org/0000-0002-1359-7948
Laura Korhonen http://orcid.org/0000-0002-1837-5930
Andreas Jud http://orcid.org/0000-0003-0135-4196

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
