## [Reviewer comments · BMJ Open]

ARTICLE DETAILS

TITLE (PROVISIONAL)	Appraisal of Published Guidelines in European Countries Addressing the Clinical Care of Childhood Sexual Abuse: Protocol for a Systematic Review.
AUTHORS	Otterman, Gabriel; Nurmatov, Ulugbek; Akhlaq, Ather; Naughton, Aideen; Kemp, Alison; Korhonen, Laura; Jud, Andreas; Vollmer Sandholm, Mary Jo; Mora-Theuer, Eva; Moultrie, Sarah; Chalumeau, Martin; Karst, Wouter; Greenbaum, Jordan

VERSION 1 – REVIEW

REVIEWER	Kellogg, Nancy The University of Texas Health Science Center at San Antonio, Pediatrics
REVIEW RETURNED	07-Jul-2022

GENERAL COMMENTS	This is an important protocol study that attempts to address the first step in providing comprehensive trauma-informed medical care for suspected victims of child sexual abuse (CSA) and in recognizing clinical presentations that should trigger a concern for sexual abuse by identifying potential gaps in national guidelines and recommendations among European nations. The authors plan to utilize the 2017 and 2019 WHO guideline and technical report as exemplary standards and plan to examine consistency between COST Action 19106 Network Countries (CANC) with guidelines meeting the criteria for inclusion in the study. The overarching goal is to improve the detection, management and care of children who may have been sexually abused. The planned literature search and assessment tools are robust. Some suggestions and questions to consider: 1. The term “guideline” refers specifically to recommendations based on a systematic review of evidence and assessment of benefits and harms of alternative care options with the goal of improved patient outcomes. However, “guideline” is often used more loosely to refer to statements of general guidance that may or may not have specific cited evidence base. Will the authors have criteria for identifying a “guideline” in their initial search? This is not mentioned in the eligibility criteria.2. The introduction discusses the importance of improving the recognition and response of health care providers (HCPs) to child sexual abuse. Comprehensive exemplary guidelines do not imply or guarantee clinician uptake. This study is the first step in assessing the quality of guidance provided to HCPs but there is an intermediate step-dissemination and implementation-that is required before clinical practice is likely to change. Perhaps this should be acknowledged.3. In the last sentence of the methods and analysis section, it states data will be summarized “using counts, frequencies,
---

	proportions and percent agreement.” It would be helpful to elaborate a bit, especially regarding “counts.” 4. In the eligibility criteria suggest editing the first bullet to “...children (<18 years) who may have experienced CSA.” 5. In the WHO technical report there are two boxes that outline the “alerting features” that should trigger consideration of CSA in the medical diagnosis. In the box specific to sexual abuse the evidence for some of these items is weak and/or age dependent (foreign bodies in vagina or anus, sexualized behavior). Additionally, the “alerting features” for “any form of maltreatment” (Box 5) is general, nonspecific and highly age dependent. Do authors plan to utilize both criteria in comparing guidelines for recognition of CSA? May wish to consider focusing on the management and care of children in whom there is already a concern for sexual abuse.
--	---

REVIEWER	Mestre i Mestre, Ruth University of Valencia, Human Rights Institute
REVIEW RETURNED	11-Oct-2022

GENERAL COMMENTS	Very interesting proposal. I suggested some ideas that will make the search and outcomes more complex, but hopefully also richer and a greater contribution to adequately responding to CSA.
--

REVIEWER	Edwards, Travonne University of Toronto, Factor-Inwentash School of Social Work
REVIEW RETURNED	22-Oct-2022

GENERAL COMMENTS	The authors are proposing to conduct a very meaningful and timely study but there are few things to consider and address prior to engaging in this systematic review. 1). You provide examples of the adverse effects of CSA but don't provide examples of what CSA is. I would include some to help the reader understand how you are constructing and defining the term as it can encompass many things. 2).When do you hope to begin this study? Given 2022 is close to ending would the search not be better positioned to include articles from Jan 1 2012 to Dec 31st 2022? 3).Can you specify the 'elsewhere components'? or remove “Training of HCPs on CSA is often lacking in Europe and elsewhere” 4) In your first research question “In comparison to the World Health Organization’s 2017 and 2019 clinical guidelines on the health sector response to child sexual abuse, what are the completeness, clarity and consistency of healthcare guidelines published by countries within the European Region addressing CSA?” Could you explain what you are referring to in regards to completeness? Is there another word that could be used? Would characteristics or components be more appropriate terms? 5) In your second research question “How do the included CANS countries guidelines compare with one another in content consistency? Are you asking what are the similarities between CANS countries guidelines? I think this question could also be
---

	bolstered by considering in ways these countries may contrast as well. 6) Will the authors who are conducting the screening undergo any inter reliability testing to assure there is some consistency? 7) It would be in the authors best interest to detail how the screening will be deciding if an article meets your inclusion criteria more explicitly. What will they be looking for in the title and abstract phase? What will they be looking for in the full text phase?
--	--

VERSION 1 – AUTHOR RESPONSE

Reviewer 1

Dr. Nancy Kellogg, The University of Texas Health Science Center at Houston Comments to the Author:

This is an important protocol study that attempts to address the first step in providing comprehensive trauma-informed medical care for suspected victims of child sexual abuse (CSA) and in recognizing clinical presentations that should trigger a concern for sexual abuse by identifying potential gaps in national guidelines and recommendations among European nations. The authors plan to utilize the 2017 and 2019 WHO guideline and technical report as exemplary standards and plan to examine consistency between COST Action 19106 Network Countries (CANC) with guidelines meeting the criteria for inclusion in the study. The overarching goal is to improve the detection, management and care of children who may have been sexually abused. The planned literature search and assessment tools are robust.

R: We thank the reviewer for affirming the robustness of the protocol methodology and recognising the overall importance of the undertaken research.

Some suggestions and questions to consider:

- 1. The term “guideline” refers specifically to recommendations based on a systematic review of evidence and assessment of benefits and harms of alternative care options with the goal of improved patient outcomes. However, “guideline” is often used more loosely to refer to statements of general guidance that may or may not have specific cited evidence base. Will the authors have criteria for identifying a “guideline” in their initial search? This is not mentioned in the eligibility criteria.**

R: We thank the reviewer for this salient point. Identification of guidelines in the initial search is broadly inclusive, based on a two-pronged search strategy (see search strategy for the database e.g., MESH terms including all available synonyms; snowball consultation with subject-matter experts in the network), and later, vetting of the publications that are identified in the search independently by two content experts prior to inclusion. Whether the authors of the included guidelines specifically cite the evidence and assess the quality of the evidence and the strength of their statements is an outcome that is being examined in the current study.

- 2. The introduction discusses the importance of improving the recognition and response of health care providers (HCPs) to child sexual abuse. Comprehensive exemplary guidelines do not imply or guarantee clinician uptake. This study is the first step in assessing the quality of guidance provided to HCPs but there is an intermediate step-dissemination and implementation-that is required before clinical practice is likely to change. Perhaps this should be acknowledged.**

R: We agree with the reviewer on this notable point. We have amended the discussion and limitations to state that our methodology does not allow us to observe what is actually being practiced, nor how the implementation of published guidelines is conducted.

- 3. In the last sentence of the methods and analysis section, it states data will be summarized “using counts, frequencies, proportions and percent agreement.” It would be helpful to elaborate a bit, especially regarding “counts.”**

R: As we undertook a standard statistical descriptive analysis with the widely-used definitions below, we felt that elaboration on this point was not necessary. We defer to the editorial team if such elaboration is recommended.

Count: the number of responses emitted during an observed period (e.g., scores: absent, partially present or absent, in combination with the proportions can be used).

Frequency: A ratio of count per observation time, often expressed as count per standard unit of time (e.g., per day, per year etc.)

- 4. In the eligibility criteria suggest editing the first bullet to “...children (<18 years) who may have experienced CSA.”**

R: We thank the reviewer for this important point with which we agree. We have edited the first bullet point in the eligibility criteria accordingly.

- 5. In the WHO technical report there are two boxes that outline the “alerting features” that should trigger consideration of CSA in the medical diagnosis. In the box specific to sexual abuse the evidence for some of these items is weak and/or age dependent (foreign bodies in vagina or anus, sexualized behavior). Additionally, the “alerting features” for “any form of maltreatment” (Box 5) is general, nonspecific and highly age dependent. Do authors plan to utilize both criteria in comparing guidelines for recognition of CSA? May wish to consider focusing on the management and care of children in whom there is already a concern for sexual abuse.**

R: We agree with the reviewer on this key observation where the exemplary reference WHO guidelines state that the evidence to support “alerting features” is weak. However, we do include these items in the data extraction template. This point will be elaborated on in the description of the data extraction methodology when the study results are prepared for publication in subsequent manuscript(s).

Reviewer 2

Dr. Ruth Mestre i Mestre, University of Valencia Comments to the Author:

Very interesting proposal. I suggested some ideas that will make the search and outcomes more complex, but hopefully also richer and a greater contribution to adequately responding to CSA.

R: We thank the reviewer for affirming that the proposal is interesting, and for the ideas and suggestions aimed to improve our research protocol.

In 2007, the Council of Europe promoted the signature of the Council of Europe Convention on Protection of Children against sexual exploitation and sexual abuse (the Lanzarote Convention). From 2010 to 2015, the One in Five campaign was run to stop sexual violence against children. The Council of Europe estimates that 1 in 5 children in Europe confronts violence, including sexual abuse. Ever since the One in Five Campaign came to an end, a European day against child sexual abuse is organized (November 18th) and many other campaigns have followed. The European Union has also worked long. In July 2020, the European Commission launched a new *EU Strategy For A More Effective Fight Against Child*

Sexual Abuse, that aims at providing a comprehensive response to the threat of child sexual abuse both, offline and online, by improving prevention, investigation and assistance to victims. This 2020 Strategy came to complement *Directive 2011/93/EU of the European Parliament and of the Council of 13 December 2011 on combating the sexual abuse and sexual exploitation of children and child pornography*, which replaced Council Framework Decision 2004/68/JHA. Eighteen years of EU efforts and normative framework do not seem to have reduced the prevalence of childhood sexual abuse in Europe. On the contrary, the Covid-19 crisis exacerbated the problem for children living with their abusers, including institutionalized children. The increase of child sexual abuse images shared online has increased as of 25% in some member states.^[1] Thus, the EU Strategy to be implemented from 2020 to 2025 comprises not only the development and enforcement of a sound legal framework to protect children and the strengthening of cooperation amongst stake holders but also the creation of a center to prevent and counter child sexual abuse, and to develop strategies to adequately protect and assist victims.

R: We thank the reviewer for providing some detailed policy developments that have transpired in the European Region in recent years. This description will be helpful in more fully providing the background when we report our study results and analysis at a subsequent stage.

Policies providing appropriate and holistic support to victims need to be yet developed and implemented, and best practices are to be shared amongst European states. Research leading to a better understanding of the problems, needs and experiences of children and adolescents that are being or have been subjected to some form of sexual abuse is much needed. In that context, the proposal of a *Systematic review of public Guidelines addressing the clinical care of childhood sexual abuse* in various European countries seems very pertinent and interesting as the study will not only identify and list guidelines in different countries but will also evaluate them by comparing their content to WHO 2017/2019 guidelines on the subject. The study will shed light on strengths and pitfalls in a very relevant area of state response to CSA (clinical care), and will provide an array of good practices, not-so-good practices, and areas of improvement. What follows are some suggestions the researchers may consider that may improve the project.

R: We thank the reviewer for endorsing the stated aims of the proposed study in which we intend to examine the characteristics of the healthcare responses to CSA in European Countries as articulated in clinical practice guidelines.

Regarding the methods of analysis, a systematic review of guidelines for health care professionals regarding child sexual abuse is proposed. Guidelines will be eligible for inclusion in the review if they meet several criteria, including to having been published between 01/01/2012 and 01/01/2022- this is, the last ten years. Yet, there's no explanation as to why those dates and not another segment. The Lanzarote Convention of 2007 set the obligation to states to assist victims- then, why not analyzing national guidelines from 2007-2020? The EU Directive 2011 established a number of obligations and standards to be followed by states, then, why not from 2011? It is true that neither the Convention nor the directive establish guidelines for HCP...

R: We thank the reviewer for noting that our protocol lacks clarity on or justification of how we limited the time period of our search. We agree with the reviewer that the Lanzarote Convention nor the 2011 EU Directive would not reasonably directly inform the publication of clinical guidelines. Our time period of about ten years was determined based on the subject matter experts' familiarity as clinicians with the publication and utilization of CSA guidelines, and the frequency of expected updates of guidelines to maintain their validity in the face of

new underlying evidence. We added text to the protocol to clarify this point and included a reference on the life span of clinical guidelines (Alderson 2013). The study period has also been extended up to November 2022, also in response to the comment by Reviewer 3 below.

...yet, if national guidelines and other reports are going to be contrasted with WHO 2017 & 2019 guidelines, it is hard to understand the inclusion for analysis of documents published before 2017. This point needs further clarification.

R: Again, we thank the reviewer for this salient comment regarding analysis of documents that were published prior to the reference standard. We note that while the WHO guidelines were first published in 2017, the underlying published evidence for the WHO guidelines has been available, largely within the span of our international electronic databases and grey literature searches.

In what regards the limitations of the study, the Study Protocol refers to the fact of addressing only national guidelines despite of the fact that healthcare in many of the analyzed countries is organized in lower administrative units. Albeit this recognition, there are no suggested strategies to lessen the limitation. The authors may consider the possibility of including in the country 'customized data extraction sheet' minimum information about the healthcare system, the existence or not of different guidelines at different administrative levels and the type of coordination amongst the different stakeholders, service providers and administrations whilst analyzing in detail only the national guidelines in the different states.

R: We thank the reviewer for suggesting an interesting, though substantial expansion of the undertaken study to include data on the organization of healthcare systems in the network. As described in the protocol, our current study emanates from the COST Action 19106 project, Multi-Sectoral Responses to Child Abuse and Neglect in Europe: Incidence and Trends (Euro-CAN). The Euro-CAN project was proposed in recognition of the complexity of multi-sectoral responses to child maltreatment, and the broader scope of Euro-CAN includes examination of the variability of health sector organization across the 34 network countries. However, this level of detail is beyond the scope of the current study.

WHO guidelines establish a series of basic principles that states must consider when drafting a comprehensive response to CSA, such as the need to decide about mandatory reporting of health care and other professionals; to ensure that through the process the best interest of the child will be respected, and his/her autonomy and participation will be guaranteed or to provide gender-sensitive care to sexually abused children. The search strategy proposed focusses on using medical subject headings, text words related to forms of childhood sexual abuse and healthcare professional guidance and it should include some of the mentioned legal/procedural aspects in order to adequately compare national guidelines to international standards.

R: In the current study, our focus is on the healthcare sector and clinical setting, including HCP access to clinical guidelines; such guidelines are anticipated to include the discussion on mandatory reporting of suspected child maltreatment by HCPs. We want to identify guidelines that a typical HCP would find if looking for clinical guidance. We are not examining more general, legal guidelines that lack a healthcare component. Some multi-sectoral dimensions, including legal aspects, are addressed in the overall Euro-CAN project, and are beyond the scope of the current study.

States should pay attention to the disproportionate vulnerability to violence of children who face multiple forms of discrimination and abuse based on their ethnicity, disability, sexual orientation, gender expression, or any other ground. According to UNICEF children with disabilities are at significantly higher risk of experiencing violence than other children, being 3.7 times more likely to experience combined forms of violence; 3.6 times more likely to be victims of physical violence and 2.9 times more likely to experience sexual abuse. Most European countries have addressed the complexity of inequality and discrimination by referring to ‘vulnerable groups’, such as migrants and asylum seekers, LGBTIQ+ people, people with disabilities, children...³The specific needs, experiences and voices of such groups have been taken into consideration both, in policies or programs addressing one group as a whole (children), and in policies and programs addressing a particular domain or problem (sexual violence). Yet too often these policies have not considered diversity within the group, nor the interactions of various forms of discrimination in the different policy domains. Thus, programs addressing violence against children may ignore the needs or experiences of girls with disabilities, because these are included in policies regarding children with disabilities or are simply ignored. The idea is that guidelines regarding the clinical care of childhood sexual abuse may be included or emanate not only in general policies addressing “violence against children” or in specific policies addressing “child sexual abuse”, but also in sectorial policies addressing the needs of children made vulnerable to violence and abuse by particular circumstances, such as institutionalized children, children with disabilities or LGBTIQ children... In that sense, the search of published guidelines and reports proposed in the study protocol should somehow include diversity within the group “children”, and diversity within policies addressing the needs of children and adolescents who have suffered some form of sexual abuse (GBV policies, cyberviolence...).

Because children facing multiple forms of discrimination not only are more vulnerable to violence and sexual abuse, but also encounter further barriers, difficulties and discrimination when seeking help, it is important to assess whether the protocols and guidelines provide tools to ensure a non-discriminatory implementation of the measures and the provision of services for children who have suffered sexual abuse.

These ideas will make both the search and outcomes more complex, but hopefully that complexity will contribute to a better understanding and development of an adequate clinical response to childhood sexual abuse that is trauma-informed, patient-centred, child and gender sensitive, and that protects all children equally, irrespective of who they are or which are their circumstances.

R: We thank the reviewer for emphasizing the crucial dimension of vulnerable children of different groups and their outsized risks of being victimised by CSA. We wholeheartedly agree with the reviewer that child participation and non-discrimination are requisite aspects that should be expected in national guidelines on CSA. We find that these points are adequately addressed in the reference WHO guidelines and refer the reviewer to the chapter on “Guiding principles derived from ethical principles and human rights standards.” We included data points on the presence of these guiding principles among the 102 items of the data extraction template that will be used to analyse the included guidelines for quality and completeness in comparison to the reference publications. The data extraction template will be published in subsequent manuscripts.

Reviewer 3

Mr. Travonne Edwards, University of Toronto Comments to the Author:

The authors are proposing to conduct a very meaningful and timely study but there are few things to consider and address prior to engaging in this systematic review.

1). You provide examples of the adverse effects of CSA but don't provide examples of what CSA is. I would include some to help the reader understand how you are constructing and defining the term as it can encompass many things.

R: We thank the reviewer for noting this lack of a proper definition of CSA. We have now added text that cites the definition of CSA by the United Nations and WHO that may be found in the glossary of the reference WHO guideline (2017) as follows:

Child sexual abuse: The involvement of a child or an adolescent in sexual activity that he or she does not fully comprehend and is unable to give informed consent to, or for which the child or adolescent is not developmentally prepared and cannot give consent, or that violates the laws or social taboos of society. Children can be sexually abused by both adults and other children who are – by virtue of their age or stage of development – in a position of responsibility or trust or power over the victim. It includes incest which involves abuse by a family member or close relative. Sexual abuse involves the intent to gratify or satisfy the needs of the perpetrator or another third party including that of seeking power over the child (3). Adolescents may also experience sexual abuse at the hands of their peers, including in the context of dating or intimate relationships. Three types of child sexual abuse are often distinguished: (i) non-contact sexual abuse (e.g., threats of sexual abuse, verbal sexual harassment, sexual solicitation, indecent exposure, exposing the child to pornography); (ii) contact sexual abuse involving sexual intercourse (i.e., sexual assault or rape – see below); and (iii) contact sexual abuse excluding sexual intercourse but involving other acts such as inappropriate touching, fondling and kissing. Child sexual abuse is often carried out without physical force, but rather with manipulation (e.g., psychological, emotional or material). It may occur on a frequent basis over weeks or even years, as repeated episodes

2). When do you hope to begin this study? Given 2022 is close to ending would the search not be better positioned to include articles from Jan 1 2012 to Dec 31st 2022?

R: Our systematic review commenced in May 2022. We agree with the reviewer that the search should be expanded through the ongoing year, and we are conducting searches to include data up to November 2022. This point has been updated in the Methods section.

3). Can you specify the 'elsewhere components'? or remove "Training of HCPs on CSA is often lacking in Europe and elsewhere"

R: We agree with the reviewer on this point and have removed reference to the 'elsewhere' component.

4) In your first research question "In comparison to the World Health Organization's 2017 and 2019 clinical guidelines on the health sector response to child sexual abuse, what are the completeness, clarity and consistency of healthcare guidelines published by countries within the European Region addressing CSA?" Could you explain what you are referring to in regards to completeness? Is there another word that could be used? Would characteristics or components be more appropriate terms?

R: By completeness, we mean the state or condition of having all the necessary or appropriate components, which is a measure of the quality of published guidelines. This is an accepted term of art in clinical guideline evaluation (see e.g., Blangis 2021).

5) In your second research question “How do the included CANC countries guidelines compare with one another in content consistency? Are you asking what are the similarities between CANC countries guidelines? I think this question could also be bolstered by considering in ways these countries may contrast as well.

R: We thank the reviewer for this comment. We have added text to the second research question to explicitly state that we aim to identify similarities, i.e., consistencies, but also to identify any manifest discrepancies between guidelines issued in European countries.

6) Will the authors who are conducting the screening undergo any inter reliability testing to assure there is some consistency?

R: We thank the reviewer for querying for interrater reliability (IRR) testing. The IRR test can be used; however, it has widely recognized limitations. Due to these limitations, and in an effort to address introduction of bias in our analysis, we opted for the following strategy that we argue has become standard methodology in conducting SRs: "two reviewers will screen, identify, extract data, appraise critically and reach a consensus between them. If there is some disagreement, a third reviewer will arbitrate."

7) It would be in the authors best interest to detail how the screening will be deciding if an article meets your inclusion criteria more explicitly. What will they be looking for in the title and abstract phase? What will they be looking for in the full text phase?

R: Screening is based on widely-used PICOS methodology. We have articulated inclusion and exclusion criteria. In the title and abstract phase, if a paper clearly meets these criteria, we will include it. In some cases, it may be difficult for investigators to judge eligibility criteria, which is why we leave papers for which there is uncertainty for the next stage, the full text phase screening. At that stage, at least two investigators will make a final decision on inclusion or exclusion based on our strict criteria and, if needed, with arbitration by a third investigator.

Please note: In addition to the above responses to comments, the authors made some minor corrections to the manuscript in formatting and formulations that are apparent in the tracked version.

In summary, my co-authors and I are grateful for the Editor and reviewers' comments and advice, which have helped us to edit and submit a revision of our manuscript for your consideration. We trust that the above responses are to your satisfaction and that the manuscript has improved. Please do not hesitate to contact me should you require any further information.

We look forward to the Editorial Office's response in due course.

VERSION 2 – REVIEW

REVIEWER	Kellogg, Nancy The University of Texas Health Science Center at San Antonio, Pediatrics
REVIEW RETURNED	07-Dec-2022

GENERAL COMMENTS	The authors plan to utilize the 2017 WHO guideline and 2019 WHO technical report as exemplary standards and plan to examine consistency between the WHO recommendations and guidelines published by COST Action 19106 Network Countries (CANC) that meet the criteria for inclusion in the study. The overarching goal is to improve the detection, management and care of children who may have been sexually abused. The planned literature search and assessment tools are robust. Some suggestions and questions to consider:  1. Based on the systematic review, the WHO derived good practice statements and recommendations. Although the authors state that they anticipate encountering "guidelines" with systematic reviews among CANC publications, it may be more efficacious to compare the good practice statements from the WHO documents and the protocols/good practice statements identified within CANC. Suggest changing title to "Appraisal of published practice statements (or guidance)...." for semantic clarity. 2. This manuscript references the 2017 WHO guidelines on "Responding to children and adolescents who have been sexually abused" and sections of the 2019 WHO guidelines that discuss identification and prevention of recurrence of CSA as the standards against which the CANC documents will be compared. However, it seems unlikely that individual CANC guidelines/protocols will include CSA recognition, response and prevention together (just as the WHO documents are also separate for response vs recognition and prevention). The response to suspected sexual abuse typically follows a child's disclosure (which tends to occur outside the healthcare setting) and applies to specialty programs whereas recognition algorithms pertain to clinicians in acute and primary health care settings. This is in contrast, for example, to recognition and response to physical abuse and neglect which are generally addressed concurrently across a variety of health care settings. For these reasons, it may make more sense to focus on (or at least separate out) "response" to CSA which would entail comparison of good practice/protocols at hospitals and clinics that specialize in assessing suspected child sexual abuse. "Recognition" is important, but it is unclear whether there are published protocols/good practices/guidelines that address screening for sexual abuse based on case-finding (preferred to general screening, as per 2019 WHO recommendations/good practice statements). As the authors mention, some of the "red flags" for sexual abuse described in the 2019 WHO technical report do not have a good evidence base. 3. The authors indicate they plan to utilize the section on "prevention of recurrence of CSA" in the 2019 WHO guidelines. This section references studies "looking at psychosocial and pharmacological interventions to decrease recurrence of child maltreatment." For sexual abuse, studies of interventions with adolescent and adult sex offenders were reviewed and WHO provided "no recommendations" because there was no "sufficient evidence on the effectiveness (or harms) of psychosocial and/or pharmacological intervention for the reduction of recurrence of child maltreatment." It is not clear how the authors plan to compare
--

	this WHO finding of “no recommendation” for prevention of recurrent CSA with the CANC guidelines in regards to prevention.
--	--

REVIEWER	Mestre i Mestre, Ruth University of Valencia, Human Rights Institute
REVIEW RETURNED	19-Dec-2022

GENERAL COMMENTS	Research leading to a better understanding of the problems, needs and experiences of children and adolescents that are being or have been subjected to some form of sexual abuse is much needed. Policies providing appropriate and holistic support to victims need to be yet developed and implemented, and best practices are to be shared amongst European states. In that context, the proposal of a Systematic review of public Guidelines addressing the clinical care of childhood sexual abuse in various European countries is both, important and pertinent. The proposed study will list existing guidelines in different countries and compare their content to WHO 2017/2019 guidelines, shedding light on strengths and pitfalls in a very particular and relevant area of state response to CSA, which is clinical care. Although the project is consistent and interesting, the researchers may want to consider some of my suggestions. 1. I had previously indicated that a clarification as to the period to be analyzed was needed, because state guidelines are going to be confronted to WHO 2017-2019 guidelines. The January 2012/November 2022 period was decided taking into consideration (1) experts’ familiarity with the publication and utilization of CSA guidelines; (2) the estimated frequency of updates of guidelines required to maintain their validity (60 months); and (3) the fact that reports issued prior to 2012 may be outdated. Although those criteria are relevant, it seems awkward to me to confront 2012 state guidelines to 2017/2019 WHO guidelines to evaluate state’s compliance with the international standards. It may be the case that 2012 is an excellent timing, but it needs to be explained. The Directive 2011/93/EU of the European Parliament and of the Council of 13 December 2011 on combating the sexual abuse and sexual exploitation of children and child pornography, which replaced Council Framework Decision 2004/68/JHA, could provide a reason for examining documents published before 2017, but the authors do not clarify it. 2. WHO guidelines establish some general principles and standards for responding to CSA, such as the need to ensure the best interest of the child throughout the process; to guarantee the autonomy and participation of the victim, or to provide gender-sensitive care to sexually abused children. Most importantly, WHO guidelines require states to decide about mandatory reporting of HCP. Accordingly, the search strategy should include, at least, whether it is mandatory or not for HCP to report CSA to the authorities. This is important because according to the Council of Europe’s estimates, 90% of cases of CSA are not reported to the police and because reporting can be a source of stress and fear for children and families. 3. Recent Council of Europe campaigns on sexual abuse of children remind us that it can happen anywhere (at home, in school, online...) and in most cases (70 to 85%) it is inflicted by someone the child knows. These numbers tell little about circumstances that may make some children more vulnerable to violence, including sexual violence and abuse. According to
--

	UNICEF children with disabilities are at significantly higher risk of experiencing violence than other children, being 3.7 times more likely to experience combined forms of violence; 3.6 times more likely to be victims of physical violence and 2.9 times more likely to experience sexual abuse. Most European countries have addressed the complexity of inequality and discrimination by referring to ‘vulnerable groups’, such as migrants and asylum seekers, LGBTIQ+ people, people with disabilities, children... The specific needs, experiences and voices of such groups may have been taken into consideration either in policies or programs addressing one group as a whole (children), or in policies and programs addressing a particular domain or problem (sexual violence). Yet too often these policies have not considered diversity within the group, nor the interactions of various forms of discrimination in the different policy domains. Thus, programs addressing violence against children may ignore the needs or experiences of girls with disabilities or may include them in policies regarding children with disabilities. This suggest that guidelines regarding the clinical care of childhood sexual abuse may be included not only in general policies addressing “violence against children” or in specific policies addressing “child sexual abuse”, but also in sectorial policies addressing the needs of children made vulnerable to violence and abuse by particular circumstances, such as institutionalized children, children with disabilities or LGBTIQ children... The proposed search of published guidelines and reports should somehow include and address diversity within the group “children”, and diversity within policies addressing the needs of children and adolescents who have suffered some form of sexual abuse (GBV policies, cyberviolence...). For instance, the analysis could include policies regarding children with disabilities; clinical care guidelines for children with disabilities; or display information as to whether or not all children are included in a particular state guidelines. Hopefully these suggestions can contribute to the study in the provision of a better understanding and development of an adequate clinical response to childhood sexual abuse that is trauma-informed, patient-centered, child and gender sensitive, and that protects all children equally, irrespective of who they are or which are their circumstances. I understand that some of them may seem to miss the core of the study (clinical care guidelines) but a comprehensive response to CSA may demand from us to look out of the boxes. 1 UNICEF, Th estate of the world’s Children: Children with disabilities, United Nations publication, Sales N° E.13.XX.1, p44. 2 Sosa, L. and R. Mestre: Ensuring the non-discriminatory implementation of measures against violence against women and domestic violence: article 4, paragraph 3 of the Istanbul Convention, Council of Europe, 2022.
--	--

VERSION 2 – AUTHOR RESPONSE

Responses to the individual comments for Reviewer 1:

Dr. Nancy Kellogg, The University of Texas Health Science Center at Houston

Comments to the Author:

The authors plan to utilize the 2017 WHO guideline and 2019 WHO technical report as exemplary standards and plan to examine consistency between the WHO recommendations and guidelines published by COST Action 19106 Network Countries (CANC) that meet the criteria for inclusion in the study. The overarching goal is to improve the detection, management and care of children who may have been sexually abused. The planned literature search and assessment tools are robust. Some suggestions and questions to consider:

1. Based on the systematic review, the WHO derived good practice statements and recommendations. Although the authors state that they anticipate encountering "guidelines" with systematic reviews among CANC publications, it may be more efficacious to compare the good practice statements from the WHO documents and the protocols/good practice statements identified within CANC. Suggest changing title to "Appraisal of published practice statements (or guidance)" for semantic clarity.

We thank the reviewer for noting that the reference WHO guidelines also include good practice statements which are a type of recommendation that is stated even with the lack of evidence to support recommendations. Our data extraction template will examine whether both the evidence-based recommendations are included and when good practice statements are identified. We have added a sentence to the Introduction for clarification, as well as a phrase on this in the Eligibility Criteria, and note that the Data items already indicated that we would be comparing with the reference good practice statements.

Good practice statements can be considered a type of recommendation within a clinical practice guideline, though not all clinical practice guidelines will include good practice statements (Lofti et al. below). We therefore do not find it necessary to amend the title of the protocol manuscript, though we plan to elaborate on this important point regarding the nature of practice statements embedded in guidelines when we present and discuss the results of our analysis in a subsequent publication.

Lotfi T, Hajizadeh A, Moja L, Akl EA, Piggott

T, Kredo T, Langendam MW, Iorio A, Klugar M, Klugarová J, Neumann I. A taxonomy and framework for identifying and developing actionable statements in guidelines suggests avoiding informal recommendations. *Journal of clinical epidemiology*. 2022 Jan 1;141:161-71.

VERSION 3 – REVIEW

REVIEWER	Kellogg, Nancy The University of Texas Health Science Center at San Antonio, Pediatrics
REVIEW RETURNED	07-Mar-2023

GENERAL COMMENTS	Reviewer concerns were adequately addressed in this revision of the manuscript. A remaining suggestion is to ensure consistency throughout the manuscript regarding the intent to examine guidelines/good practice statements that encompass the recognition of, AND response to, child sexual abuse. As previously discussed, these are distinct issues involving distinctly different groups of HCPs and health care facilities. For example:  1. In the first Objective, mention is made of comparing guidelines that encompass the response to CSA but not the recognition of CSA 2. Data synthesis section mentions mentions comparison of guidelines that refer to responding to CSA, but not the recognition of CSA.
---

REVIEWER	Mestre i Mestre, Ruth University of Valencia, Human Rights Institute
REVIEW RETURNED	08-Mar-2023

GENERAL COMMENTS	I enjoyed reading the authors' comments to the reviewers' suggestions and I think this dialogue improved the proposal, which was very interesting and needed from start. Looking forward to reading the "derivates" of it.
--

VERSION 3 – AUTHOR RESPONSE

Responses to the individual comments for Reviewer 1:

Dr. Nancy Kellogg, The University of Texas Health Science Center at Houston
Comments to the Author:

Reviewer concerns were adequately addressed in this revision of the manuscript.
We thank the reviewer for following the revisions with suggestions that allow us to improve the manuscript for publication.

A remaining suggestion is to ensure consistency throughout the manuscript regarding the intent to examine guidelines/good practice statements that encompass the recognition of, AND response to, child sexual abuse. As previously discussed, these are distinct issues involving distinctly different groups of HCPs and health care facilities.

For example:

1. In the first Objective, mention is made of comparing guidelines that encompass the response to CSA but not the recognition of CSA
2. Data synthesis section mentions mentions comparison of guidelines that refer to responding to CSA, but not the recognition of CSA.

We have added reference to recognition of and response to CSA in both the locations in the manuscript noted by the reviewer as well as in the Rationale section of the manuscript. As we complete the data analysis of our systematic review and prepare to present our findings, we intend to diligently address this important point of discussion of the distinct purposes and scope of CSA guidelines.

Reviewer: 2

Dr. Ruth Mestre i Mestre, University of Valencia

Comments to the Author:

I enjoyed reading the authors' comments to the reviewers' suggestions and I think this dialogue improved the proposal, which was very interesting and needed from start. Looking forward to reading the "derivates" of it.

We thank the reviewer for following the revisions and engaging in dialogue which we agree has approved our current protocol manuscript and will assist us further as we look toward data analysis and presentation of our results in subsequent publications.